# Taxonomic distinctness and diversity patterns of a polychaete (Annelida) community on the continental shelf of the Southern Gulf of Mexico

Benjamín Quiroz-Martínez[1]*, Pablo Hernández-Alcántara[1], David Alberto Salas de León[1], Vivianne Solís-Weiss[2], María Adela Monreal Gómez[1], León Felipe Álvarez Sánchez[3]

1 Unidad Académica de Ecología y Biodiversidad Acuática, Instituto de Ciencias del Mar y Limnología, Universidad Nacional Autónoma de México, Ciudad de México, México, 2 Unidad Académica de Sistemas Arrecifales, Instituto de Ciencias del Mar y Limnología, Universidad Nacional Autónoma de México, Puerto Morelos, Quintana Roo, México, 3 Unidad de Informática Marina, Instituto de Ciencias del Mar y Limnología, Universidad Nacional Autónoma de México, Ciudad de México, México

* bquirozm@cmarl.unam.mx

## Abstract

The spatial patterns of taxonomic diversity of annelid polychaete species from the continental shelf in the Southern Gulf of Mexico were examined in this study. We used taxonomic distinctness and its spatial variations to explore the diversity patterns and how they change between Southern Gulf of Mexico regions. In addition, using taxonomic distinctness as a dissimilarity measure and Ward's Clustering, we characterized three distinct faunal assemblages. We also investigated patterns of richness, taxonomic distinctness, and distance decay of similarity between sampling stations as a ß-diversity measure. Finally, we examined the spatial relationships between polychaete assemblages and environmental variables to test the relative importance of spatial and environmental components in annelid polychaete community structure from the Southern Gulf of Mexico. We used a combination of eigenvector-based multivariate analyses (dbMEMs) and distance-based redundancy analysis (dbRDA) to quantify the relative importance of these explanatory variables on the spatial variations of taxonomic distinctness. The significance level of spatial and environmental components to the distribution of polychaete species showed that the combined effect of spatial processes and sediment characteristics explained a higher percentage of the variance than those parameters could alone.

## 1. Introduction

Explaining the causes of biodiversity variability at various scales and correlating these variations to changes in environmental conditions are two of ecology's primary goals [1, 2]. To estimate how biodiversity may respond to climate change, it is essential to understand the

**Data Availability Statement:** Data is available from Seanoe digital repository (https://doi.org/10.17882/98735).

**Funding:** This research was funded by the Universidad Nacional Autónoma de México DGAPA/PAPIIT Project IA202321 "Análisis y visualización de bases de datos de biodiversidad en sistemas marinos para determinar cambios en los patrones de distribución de la riqueza de especies y la abundancia de la fauna marina del Golfo de México" and through grant 628 from the Instituto de Ciencias del Mar y Limnología, Universidad Nacional Autónoma de México. The funders had no role in study design, data collection and analysis, decision to publish, or preparation of the manuscript.

**Competing interests:** The authors have declared that no competing interests exist.

patterns of species distribution and the ecological and evolutionary processes that lead to these patterns. Abiotic factors influence the borders of the species ranges based on their physiological preferences [3]. Numerous complex mechanisms have determined the distribution of species at diverse geographical and temporal scales [3].

Over the past few years, there has been a remarkable growth in research on marine biodiversity patterns, and understanding the spatial distribution of species in soft-bottom habitats is essential to comprehend how species interact with one another and with the environment [4] and how benthic communities and ecosystems function [5, 6]. Community ecology studies frequently address the connection between species composition and environmental factors [7–12]. Spatially structured environmental variation plays a vital role in species distribution [13]; consequently, discriminating spatial ecological structures in species composition, distribution, and diversity is a pivotal question in community ecology [14]. However, the spatial scale can also incorporate confounding factors thus, statistical models should include them as predictors or covariable parameters [15]. The local scale is generally considered the spatial autocorrelation created by community dynamics, while the regional scale corresponds to the scale of the environmental drivers [13].

The spatial component in community analysis has been incorporated using a variety of methodologies [16, 17]. For example, ordination analysis [18, 19] can partition the variance across different variables. This approach entails assessing environmental variables, creating a spatial coordinates matrix, and characterizing these elements' individual and combined contributions to the community structure. Due to its ability to explain the regional and local dynamics that shape communities, variation partitioning has become a crucial exploratory tool [5, 20, 21]. Processes associated with sediment properties, temperature, salinity, dissolved oxygen, and nutrient content are among the drivers that structure the geographical distribution of biodiversity [12, 22, 23].

Annelid polychaetes are among the most diverse and characteristic groups of benthic macroinvertebrates [24, 25] and one of the richest concerning species numbers [26, 27]; it includes nearly 11,500 accepted nominal species and is often the dominant group in benthic macrofauna [28], representing up to 60% of the species and over 70% of the abundance, mainly in soft-bottom communities [24, 29]. They are ubiquitous in soft sediments and show significant spatial distribution patterns under the influence of environmental gradients [30]. Polychaetes can be used as "indicators" of different ecological conditions [31, 32], and they can be particularly informative in assessing the health of benthic environments. Therefore, the analysis of the diversity patterns of this fauna can be crucial in understanding the whole ecosystem functioning since the biological processes that determine these patterns can reflect those of the entire ecosystem [33–39]. On the continental shelf of the Gulf of Mexico, the polychaetes have been the focus of comprehensive research for decades as they represent a crucial group in abundance and diversity [40]. However, most of these studies only present taxonomic lists and are often fragmented or isolated, so information about their distribution patterns related to environmental factors is still scarce. A comprehensive polychaete database was therefore built using the results of several research projects examining the occurrence of this group in the continental shelf of the Southern Gulf of Mexico.

Using distance-based redundancy analysis (dbRDA) and variation partitioning, this study aims to analyze the environmental factors that influence polychaete diversity at small and intermediate scales along the continental shelf of the Southern Gulf of Mexico. We assess the relative impact of spatial and environmental components on polychaete community structure and examine the spatial relationships between polychaete assemblages and environmental variables.

## 2. Materials and methods

### 2.1. Study area

The Gulf of Mexico is a marginal sea shared by three countries, Mexico, Cuba, and the United States of America. It is recognized as a large marine ecosystem [41], where activities with significant economic importance, such as fisheries, transportation, and gas and oil extraction are constantly increasing [41]. The Southern Gulf of Mexico (18° 30′22° 20′ N; 89° 41′–97° 4′ W), delimited by an imaginary line that goes from the port of Tampico in the State of Tamaulipas to Progreso in the State of Yucatan, covers approximately 46 000 km$^2$. Its southeastern portion (Fig 1, upper panel) is composed of two distinct regions: The Campeche Bank, an expansive shelf characterized by a gentle slope and uneven bottoms and sandbanks, coral reefs, and native biogenic and autogenic sediments along most of the Yucatan Peninsula's coast, and the Bay of Campeche, which is a narrow shelf with its upper limit at 130 m depth and between 45 and 65 km wide (Fig 1) [42, 43]. Three seasons characterize the southern Gulf of Mexico, the dry season from March to May, the rainy season from June to October, and the winter storms season, locally called the "Nortes" season, from November to February [44]. The southern Gulf of Mexico has combined influences from river run-off to the west and winter storms ("Nortes") [45, 46]. The study area has been classified according to the sediments' carbonate content: the "terrigenous" region to the west with less than 25% and the "carbonate" region to the east and north, with more than 75% with a transitional carbonate-terrigenous region, between the first two, whose limits change throughout the year, with intermediate values [47–49]. The terrigenous sediments are predominantly silty with variable amounts of gravel, sand, and clay, while fragments of shells, corals, and algae mainly constitute the carbonate ones. The Southern Gulf of Mexico is a complex tropical domain with intense river discharges to the west from the Grijalva-Usumacinta River system; it modifies the temperature and salinity, inducing coastal frontogenetic processes, as well as being the primary source of terrigenous sediments [49, 50]. Discharges from the Grijalva-Usumacinta and San Pedro-San Pablo rivers, "Nortes", summer rains, and intrusion of oceanic water from the Caribbean Sea are the physical driving processes on the continental shelf of the southern Gulf of Mexico [45, 50]. The Caribbean current drives the water circulation pattern in the spring and summer, with a south-southwest flow however, in the autumn and winter, the flow shifts to an east-northeast direction [51–53]. As the Caribbean Current flows northward through the Yucatan Channel, it is renamed the Yucatan Current. The Caribbean Tropical Surface Water (CTSW: T>28°C and S>36.4) and the Caribbean Subtropical Underwater (CSUW: 22.2<T<26°C and 36.4<S<36.7) influence the Campeche Bank since the Yucatan current flows into the Gulf, not only contributing significantly to the Loop Current, but another branch flows westward on the Yucatan continental shelf and Campeche Bank [54].

### 2.2. Data source

The polychaete database, constructed from extensive studies carried out on the continental shelf (<200 m depth) of the Southern Gulf of Mexico (18° 30′–22° 20′ N; 89° 41′–97° 4′ W) was the primary source of information: the samples were collected during six oceanographic expeditions, on board the R/V "Justo Sierra" (Universidad Nacional Autónoma de México), from 1988 to 1996, as part of two institutional projects "IMCA" and "DINAMO" [55]. The collected specimens from 61 stations were identified to species level and deposited in the "Colección Nacional de Anélidos Poliquetos" from the Instituto de Ciencias del Mar y Limnología, Universidad Nacional Autónoma de México (CNAP-ICML, UNAM; DFE.IN.061.0598). The database includes information on all identified polychaete species, the geographical position of

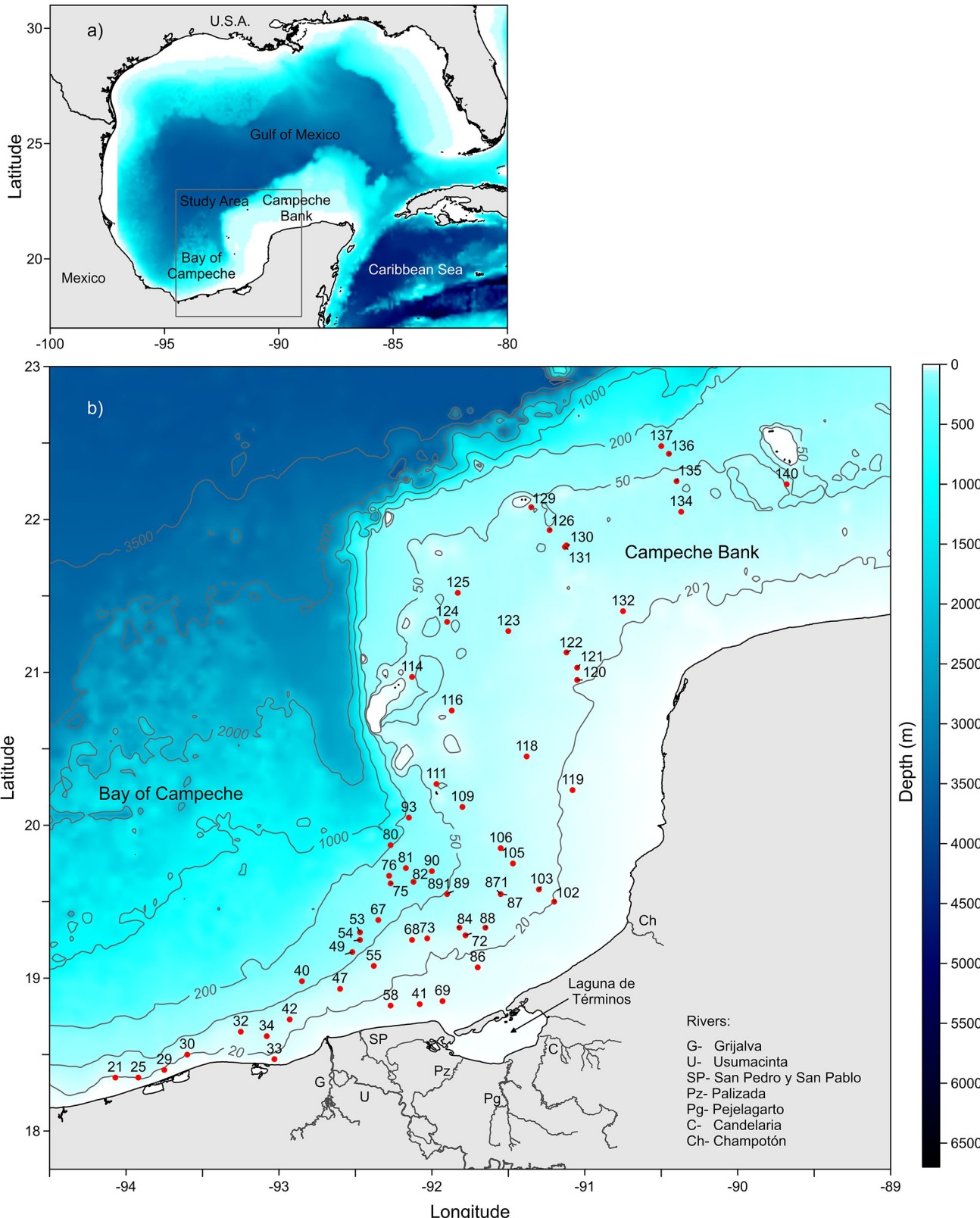

**Fig 1. Sampling stations in the Southern Gulf of Mexico.**

the sampling stations (latitude and longitude), and environmental variables such as sediment composition, depth, temperature, and salinity where they were collected. In addition, the taxonomic information was carefully checked. We verified the validity of names and synonymies and omitted species whose taxonomic identification was doubtful. We used the more recent systematic reviews and the World Polychaeta Database [56] accessed through the World Register of Marine Species (WORMS) database. Data is available from Seanoe digital repository (https://doi.org/10.17882/98735)

## 2.3. Statistical analyses

The information in the database was linked to the corresponding supra-generic levels for every species; their relationships to genus, family, and class were included. We then calculated two indices to compare polychaete diversity between sampling stations: 1) the Average taxonomic distinctness ($\Delta^+$), and 2) the variation in taxonomic distinctness ($\Lambda^+$), as defined by Warwick & Clarke (2001) and Clarke & Warwick (1998).

The Average taxonomic distinctness ($\Delta^+$) was calculated using the function:

$$\Delta^+ = [\Sigma\Sigma_{i<j}\omega_{ij}]/[S(S-1)/2] \tag{1}$$

where $\omega_{ij}$ is the taxonomic path length between species $i$ and $j$, and $S$ is the number of species).

While the variation in taxonomic distinctness ($\Lambda+$) was calculated as follows:

$$\Lambda^+ = [\Sigma\Sigma_{i<j}(\omega_{ij} - \Delta^+)]/[S(S-1)/2] \tag{2}$$

The average taxonomic distinctness ($\Delta^+$) measures the average taxonomic distance between different species in an assemblage; the higher the values of $\Delta^+$, the more significant the average taxonomic difference between species in the assemblage. Conversely, $\Lambda^+$ is the variation in branch lengths amongst all pairs of species and measures the distribution of branch lengths within a sample; the greater the variation in branch lengths, the greater the value of $\Lambda^+$.

To assess the differences in taxonomic distinctness ($\Delta^+$) from expected $\Delta^+$ values determined from the species list of polychaetes from the Gulf of Mexico, we used a randomization approach (as suggested by Clarke & Warwick [57, 58] and by Warwick & Clarke [59]). A simulated distribution was created, leading to a theoretical mean (a horizontal line displaying the taxonomic distinctness for all polychaete species shown in the graph of $\Delta^+$ or $\Lambda^+$ for all the sampling stations against richness in each station) and to a confidence funnel for each, $\Delta^+$, and $\Lambda^+$, from random subsamples of the polychaete species as suggested by Bhat and Magurran [60]. The simulation creates numerous random subsets of species, each of size $m$, from the entire species list, computes the corresponding $\Delta^+$ and $\Lambda^+$ values, and establishes a range in which 95% of these values are included. Values of $\Delta^+$ and $\Lambda^+$ located within the 95% probability funnel indicate that species diversity in the corresponding areas falls within the expected range, allowing for diversity comparisons not impacted by sample size.

To evaluate the effects of geographical distance on taxonomic composition (Distance Decay), we calculated the distance in km between all pairs of sampling stations. Next, the relationship between pairwise faunal dissimilarity and spatial distance was assessed by fitting an exponential function describing the increase in faunal dissimilarity with spatial distance [61, 62]; the R package "betapart" (functions "decay.model" and "boot.coefs") was used; as it adjusts a General Linear Model (GLM) with dissimilarity as the response variable, spatial distance as the predictor, log link, and Gaussian error. Finally, the intercept and slope parameters were bootstrapped (1000 replicates) [63, 64].

The indices were computed using the polychaete species' taxonomic hierarchy based on the Linnaean classification using the Plymouth Routines in Multivariate Ecological Research

PRIMER-E [65, 66]. The resulting matrices were examined to derive dissimilarity patterns through Ward's hierarchical cluster analysis on the taxonomic dissimilarity matrix. The optimal number of clusters was determined using the average silhouette width and the diagnostic group criteria suggested by Borcard et al. [67]. The cluster analysis was run on the $\theta^+$ taxonomic dissimilarity index, a presence/absence "beta diversity" coefficient [65, 68]. The significant differences in the assemblage structure between the two studied regions were tested with a one-way analysis of similarity (ANOSIM).

In addition, we calculated the environmental distance between each pair of stations with Euclidean distances based on all environmental variables measured. We used each sampling station's sediment composition, depth, temperature, and salinity as environmental variables. Distance-based Moran's eigenvector maps (dbMEMs), redundancy analysis (dbRDA), and variation partitioning were used to quantify the effect of spatial structure and biotic and abiotic variables on community dissimilarity (taxonomic distinctness). Before the dbRDA analysis, forward selection procedures were conducted to select the most significant spatial and environmental variables; only the dbMEMs corresponding to positive autocorrelation as spatial variables were used in the analyses [69]. Distance-based Moran eigenvector maps (dbMEMs) for the matrix of the sites' geographical coordinates were used to describe the dataset's spatial structure. The initial dbMEMs model the large-scale spatial correlation, while the last dbMEMs correspond to fine-scale spatial correlations, which may capture variation at the sampling site scale. All non-significant ($p > 0.05$) variables were eliminated from further analyses. Afterward, we performed a variation partitioning analysis to assess the relative contribution of the significant spatial and environmental variables in explaining the community dissimilarity [70]. The variation was partitioned into four fractions: (a) influence of environmental variables alone (sediment composition, depth, temperature, and salinity); (b) influence of spatial distance; (c) influence of spatially correlated variables (distance-based Moran eigenvector maps; dbMEMs); and (d) unexplained variation. The variation partitioning analysis included only the variables selected by this procedure. The significance of each fraction was tested by Monte Carlo permutation tests. In addition, the relative contribution of each significant spatial and environmental variable in determining community dissimilarity was assessed by partial dbRDA to partial out the effect of each significant environmental/spatial variable. All analyses were run in R-4.0.3 with the "vegan" and other built-in packages [71, 72]

## 3. Results

### 3.1. Taxonomic distinctness and distance decay

One hundred and seventy-three species from 40 families were recorded from the continental shelf of the Southern Gulf of Mexico. The eastern continental shelf region (Campeche Bank) had the highest number of species (164). In contrast, 125 species were collected in the western region of the study area (Bay of Campeche).

The estimated $\Delta^+$ values resulting from the polychaete species list in each of the 61 sampling stations from the Southern Gulf of Mexico showed that most of the calculated $\Delta^+$ values were close to the expected simulated mode of the funnel (Fig 2). According to these $\Delta^+$ values, most of the stations were as diverse as expected since they were located inside the funnel. However, a few sampling stations (21, 25, 29, 30, 32, 33, 34, 41, 42, 72, 73, 75, and 124), mainly located on the terrigenous shelf in the southernmost Gulf of Mexico, displayed lower diversity than expected.

The variation in taxonomic distinctness ($\Lambda^+$) showed that most stations were placed inside the funnel, close to the expected values. However, a few stations fell above the simulated funnel's upper limit, indicating higher-than-expected $\Lambda^+$ values (Fig 2).

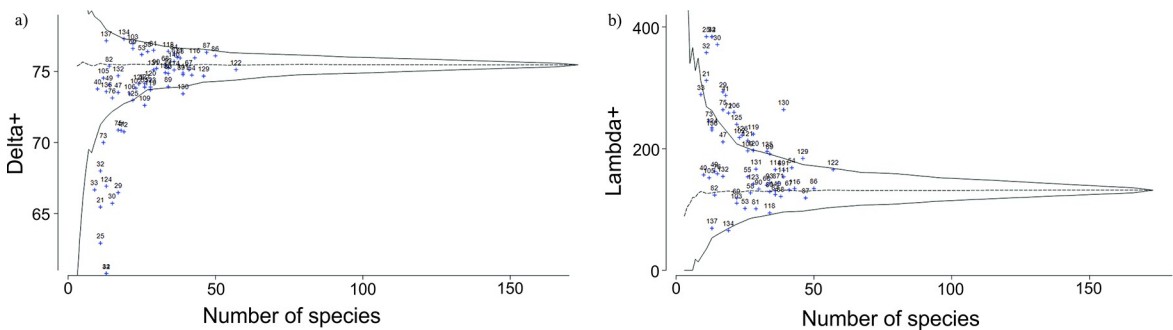

**Fig 2.** Simulated distribution of a) average taxonomic distinctness ($\Delta^+$) and b) variation in taxonomic distinctness $\Lambda^+$ (theoretical mean, horizontal, dashed line) for random subsets of species from the complete species list of 173 polychaete species from the Southern Gulf of Mexico and the 95% confidence limits (the funnel).

The exponential fit between the calculated value of $\theta^+$ taxonomic dissimilarity index as a function of the distance between all pairs of sampling stations showed an exponential relationship between compositional dissimilarity and distance. The distance among stations exhibited high variability, but most differed at less than 200 km. The dissimilarity between the polychaete taxonomic composition from all stations exponentially increased with increasing geographical distance (Fig 3).

## 3.2. Cluster analysis

Ward's Clustering, using the $\theta^+$ taxonomic dissimilarity index relating to the polychaete composition of the 61 sampling stations, showed three main faunal assemblages with high taxonomic dissimilarity (>0.75) between them (Fig 4). The ANOSIM analysis confirmed that differences between the three polychaete assemblages were significant ($R_{ANOSIM(C1-C2)} = 0.374$, $p = 0.001$; $R_{ANOSIM(C1-C3)} = 0.45$, $p = 0.001$; $R_{ANOSIM(C2-C3)} = 0.508$, $p = 0.001$). The "Terrigenous assemblage" (Cluster 1 in red) was integrated by stations located in the southernmost region of the terrigenous shelf of the Gulf of Mexico. Interestingly, sites 114, 126, and 140, associated with Cluster 1, are geographically located on the Carbonate Shelf. The "Transitional" assemblage (Cluster 2 in green) grouped the stations from the central part of the study

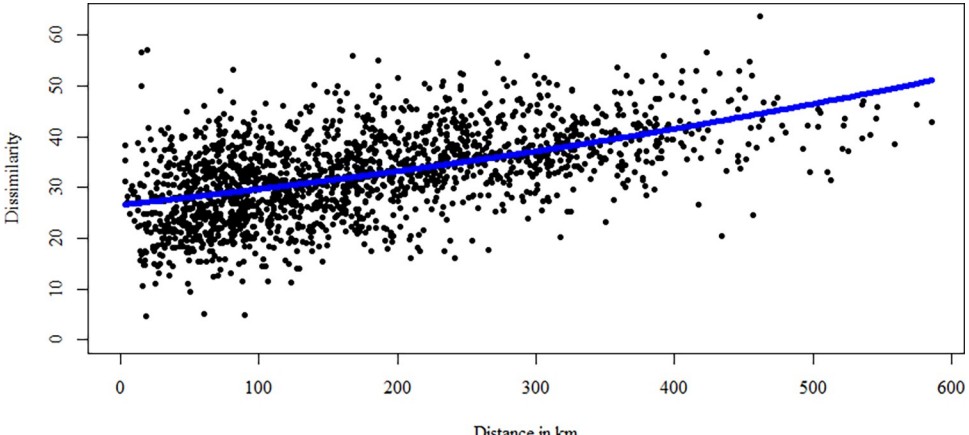

**Fig 3. Taxonomic dissimilarity index $\theta^+$ plotted against distance for all pairwise comparisons between sampling stations.** The blue line represents the exponential function describing the increase in dissimilarity (a.intercept = 26.56; b.slope = 0.001).

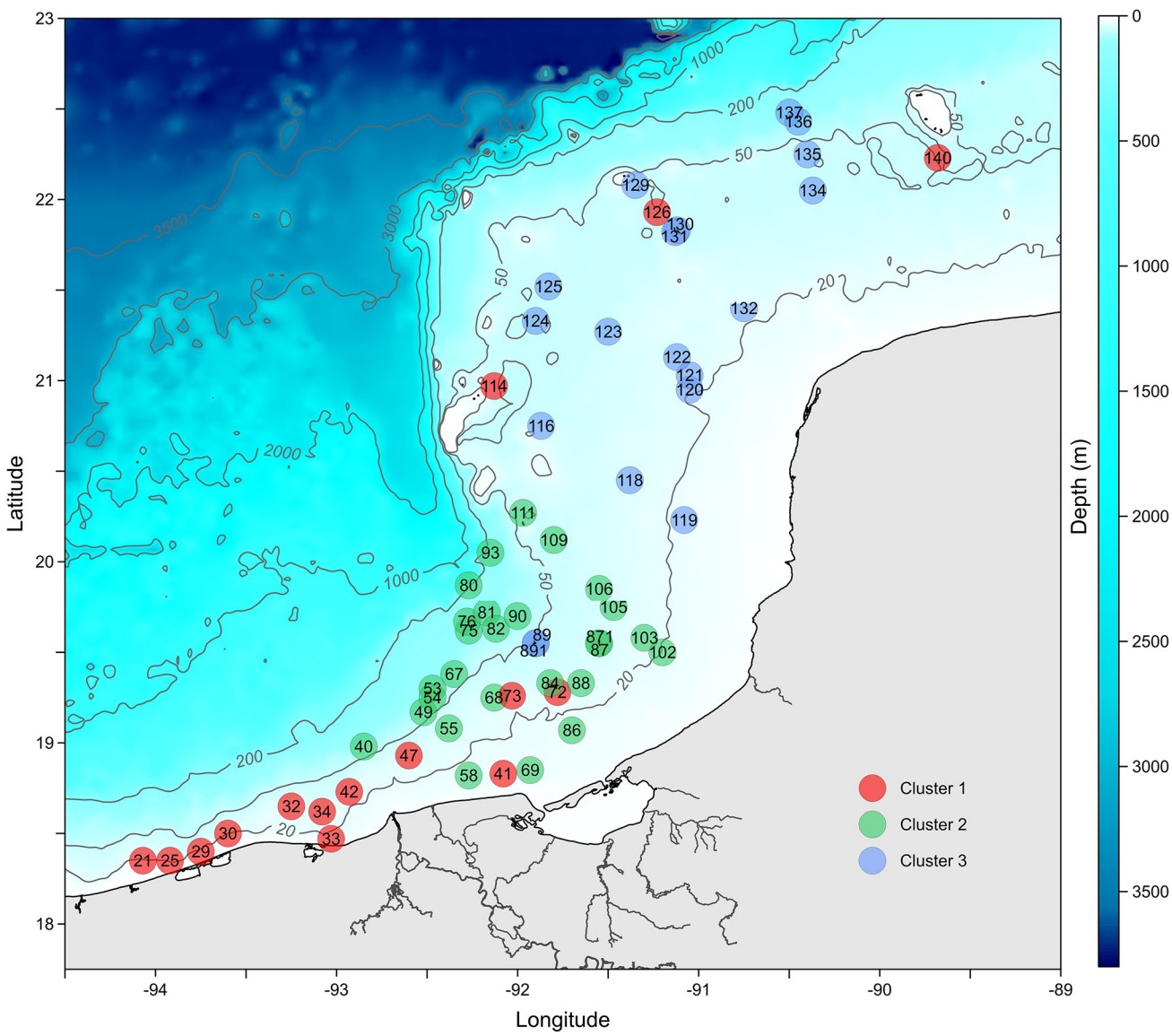

**Fig 4. Cluster analysis based on taxonomic dissimilarity and Ward's clustering method.**

area in front of Laguna de Términos, considered a sedimentary transitional region. Finally, the "Carbonated" assemblage (Cluster 3 in blue) included stations located on the carbonate shelf on the eastern Gulf of Mexico.

### 3.3. Distance-Based Redundancy Analysis (dbRDA)

The distance-based redundancy analysis (dbRDA) showed that the first axis explained 43.56% of the total variation, and the second explained 24.08% (Fig 5). The first axis represented a large-scale (with a high correlation with dbMEM1) sedimentary and thermal gradient characterized by increasing sand content and temperature. The "Carbonate" assemblage (blue) in the eastern region was associated with warm water and sandy sediments. Stations belonging to the "Terrigenous" assemblage (red) in the western study area are associated with a medium-scale (dbMEM7) salinity gradient. The "Transitional" assemblage (green) in front of Laguna de

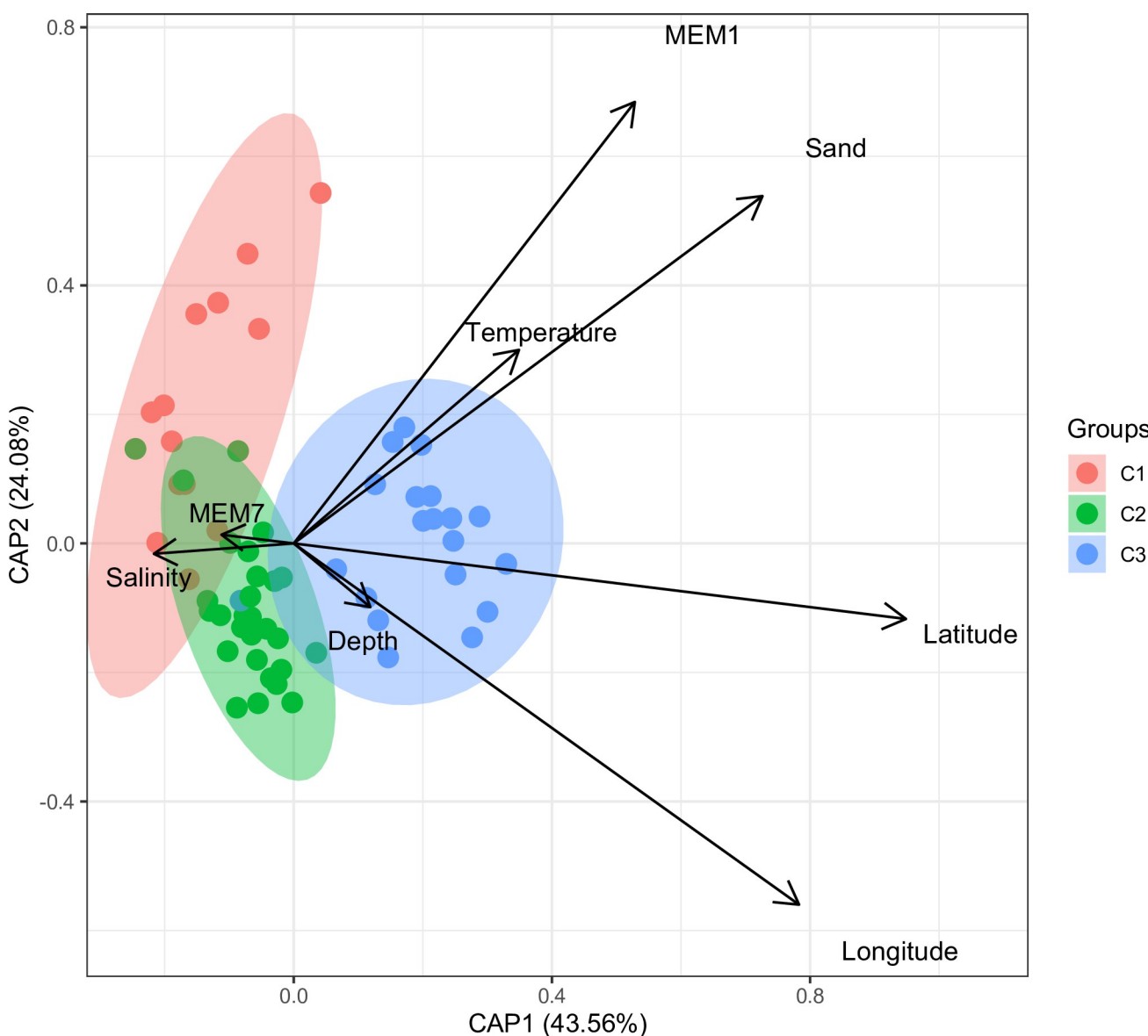

**Fig 5. Distance-based redundancy analysis (dbRDA) ordination plot showing the relationships between sampling stations based on polychaete taxonomic distinctness and spatial and environmental constraints.**

Términos was related to salinity variability and increasing mud content in the sediment (Fig 5). All the environmental variables seem to be spatially structured. The unexplained variation may be linked to describers not used in the present study or simply due to random factors.

## 3.4. Variation partitioning

The set of environmental (X1) and spatial variables (dbMEM1, X3) explains 5% of the variation in taxonomic dissimilarity across the study area. The environmental variables alone (X1 in the partitioning results) explain almost 10% of the variation, of which 5% is not spatially structured. This fraction represents species-environment relationships associated with local

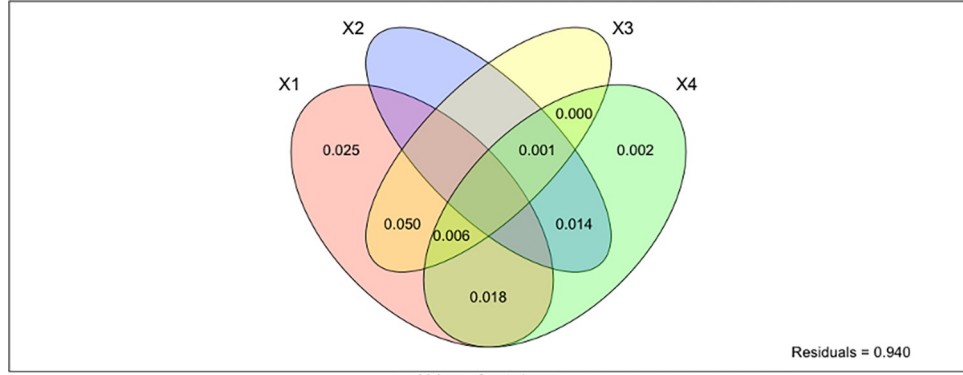

**Fig 6. Variation partitioning of the taxonomic dissimilarity into an environmental component (X1), linear spatial processes (X2), a broad-scale (X3), and fine-scale (X4) dbMEM spatial components.**

environmental conditions. Medium-scale spatial processes (dbMEM7, X4) explained 3.4% of the taxonomic variability. Finally, the linear spatial structure (x and y coordinates; X2) contributions were less significant (1.5%) (Fig 6).

## 4. Discussion

In the Southern Gulf of Mexico, intensive extraction of oil and other hydrocarbons has been carried out since the 1970s; thus, the ability to assess this industry's naturally polluting and associated anthropic impacts is important [45, 47]. Therefore, understanding the distributional patterns of the polychaetes species present can help us understand the ecological processes occurring in the benthic ecosystems and how these organisms could be used as indicators of different ecological conditions at a larger scale [12, 31].

In the Southern Gulf of Mexico, polychaete taxonomic distinctness changed across the longitudinal gradient, increasing from west to east, linked to a decline in species richness towards the terrigenous shelf in the west. The low taxonomic distinctness observed in terrigenous environments could be related to its more homogenous conditions, allowing the coexistence of taxonomically close species. In contrast, in the carbonated environment, the habitats are more varied, with highly heterogeneous bottoms having different grain sizes, enabling the occurrence of taxonomically distinct species with different ecological preferences. Sediment composition is one of the main factors influencing the settling and distribution of benthic species. Sediment type and grain size variations determine organic matter content, dissolved gases, and nutrient concentrations [73–78]. The mixed sandy-mud bottoms dominate the Campeche Bank, favoring higher densities and species richness. This contrasts with the terrigenous sediments from the western Gulf, which negatively affect the benthic communities' diversity. Our analyses allowed us to distinguish the occurrence of three distinct polychaete species assemblages with the sedimentary distribution in the southern Gulf: one located in the western region on the terrigenous shelf, another associated with the carbonate shelf to the east, and an intermediate zone, in front of Laguna de Términos.

In the Bay of Campeche, the circulation is dominated by the mesoscale cyclonic eddy, which develops in the eastern bay and moves westward, enhancing small-scale anticyclonic and cyclonic eddies [79]. In the study area, a coastal current moves, southward or south-westward, along the east coast mainly; approximately at 93°50W longitude, it converges with an eastward current along the coast to form a northward current along the shelf break [79]. The sediment distribution in the southern Gulf of Mexico (Bay of Campeche) is mainly driven by

wind-induced currents and mesoscale cyclonic eddies [79]. Coarse sediments are located near the coast in the convergence region and the core of the anticyclonic eddy. In turn, fine sediments are situated off the shelf coast and in areas where the weakest currents are recorded [79]. Carbonates dominate sediments on the eastern side of the Bay of Campeche since no river discharges exist or any other systems that could contribute terrigenous sediments, whereas, in the southernmost part of the bay, where major rivers are located, the bottom sediments are mainly of terrigenous origin [79].

The composition and structure of benthic assemblages can be influenced by numerous physical and biological factors [80, 81] that define their zonation along environmental gradients [82]. In the study area, sediment variations were essential in structuring the polychaete assemblages. In addition, bathymetric fluctuations usually promote changes in other depth-dependent physical variables [77, 83–85]. Particularly in the Southern Gulf, the distance between the sampling stations and their geographical proximity directly affected the faunal differences since the dissimilarity in polychaete species composition was a function of the distance separating them. Graco-Roza et al [86] have proposed that taxonomic distance decay is a useful tool for many aspects of biogeographical research because it reflects dispersal-related factors in addition to species responses to spatially structured environmental variables. Distance decay curves at regional or local scales are primarily shaped by the organisms' biological characteristics (niche requirements, dispersal ability) and environmental interactions [87]. Nearby locations should typically have highly different species compositions when a species has a narrow geographic range, and power-law or negative exponential functions can be used to approximate distance-decay curves. However, close-by biological communities may be very similar when species have large spatial ranges relative to the study extent. This leads to a low or zero rate of change in community composition at the closest distances, while the actual decrease in community similarity would start at intermediate distances [87]. Several studies have shown that sediment composition can be crucial in controlling benthic communities in marine environments [12, 80, 88]. Granados-Barba (2001) and Quiroz-Martínez et al. [12, 89] also showed that, in the southern Gulf, the polychaete species richness increased with depth.

In this sense, our results, based on distance-based Redundancy analysis, showed that sediment type (and the dbMEMs) were the primary factor, rather than depth, to determine the polychaete faunal structure in the Southern Gulf of Mexico. For example, the syllids, usually small and highly motile, constituted the most diverse polychaete family and were mainly found in carbonated bottoms, where coarse sands mixed with coral and shell rubble dominate. However, the syllids become uncommon in mud and sandy-mud terrigenous sediments. In the soft bottoms of the eastern Campeche Bay, syllids species richness generally followed a decreasing gradient from carbonate to terrigenous sediments (east to west) [90]. In addition, dbMEMs spatial processes explained a higher proportion of the variance in polychaete community structure than the environmental variables. Although polychaete larvae can disperse long distances, adult dispersal is often restricted and occurs over local/small spatial scales, mainly in intertidal areas and during local disturbance events [5]. Our results suggest that species composition was spatially correlated at shorter distances and that dispersal limitation and species response to spatially structured environmental gradients might be involved in determining polychaete distribution patterns.

The observed changes in sediment type affecting the establishment and development of benthic organisms can be associated with the water-sediment interphase dynamics that modify the organisms' excavation processes and limit the number of species that can settle [91]. According to Santibañez-Aguascalientes et al. [92], the carbonate bottoms to the east comprise medium and fine calcarenites and oxides that provide suitable habitats for benthic communities [49, 90, 93]. In turn, the terrigenous bottoms are influenced by fluvial discharges dragging

mud and sand sediments and harbor a high diversity of polychaete species adapted to turbid bottoms and terrigenous material [80, 94].

Characterizing distinct groups of invertebrates can be a complex process since, along biological gradients, a gradual transition between different assemblages frequently occurs [77, 83]. In the present study, the separation of species groups based on multivariate analysis and taxonomic distinctness allowed us to distinguish three large polychaete assemblages; this indicates a gradual change in the composition of the polychaete fauna along the longitudinal gradient. The use of taxonomic distinctness (which includes the taxonomic relationships between species in an assemblage) and Ward's Clustering allowed for a more detailed characterization of the polychaete fauna from the Southern Gulf of Mexico. The present results update and refine, adding a third assemblage, the results previously described by Quiroz-Martínez et al. [12]. Furthermore, polychaete assemblages distributed in the transitional region were different from the other sedimentary environments; this could be related to the high environmental variability of the transitional area, characterized by the transition from terrigenous to carbonated sediments, modulated by the Grijalva-Usumacinta plume and the oil extraction activities that affect the benthic fauna by reducing species diversity and richness [79, 95, 96]. Ecological research in the Southern Gulf of Mexico shows that "Nortes" and summer rains provoking high river run-off are the most important physical processes influencing large-scale benthic community structure across the carbonate and terrigenous shelves [45, 47, 90, 97–99].

Finally, connectivity is another process that could explain the results presented in this study. Connectivity is an essential process in any marine ecosystem because it influences, among other things, the colonization of habitats, the maintenance of environmental services, and the persistence and survival of marine benthic populations [100]. According to Treml et al. [101], there are 4-stage processes that explain the connectivity between populations; it includes the dispersion, transport, and movement of the organisms, which is strongly determined by the advection potential, which in turn is related to the magnitude of the currents. This could explain the three distinct faunal assemblages observed in this study, considering that the Southern Gulf of Mexico is highly dynamic, with the confluence of mesoscale physical processes and intense coastal currents, which could have influenced the polychaete patterns of distribution presented in this study. For example, the connectivity between coral reef systems from the Gulf of Mexico has already been demonstrated using biological data [102–106]. In addition, Salas-Monreal et al. [52] elucidated the different pathways that connect the Western Gulf of Mexico reef systems using monthly geostrophic velocities, sea surface temperature, and chlorophyll-*a* values. They found high connectivity between the reef systems that bring substrates, suspended matter, and organisms from the Campeche Reef System (CRS) in the Campeche Bank to the Veracruz Reef System (VRS) in the Southwestern Gulf of Mexico. According to them, the geostrophic current velocities suggested various paths connecting the various coral reef habitats, with two open ocean pathways in addition to the traditional coastal alongshore path, which is constrained to the continental shelf, with the first linking the Campeche Reef System with the Veracruz System. The water circulation pattern is driven by the Caribbean current during the spring and summer, with south to southwestern direction; however, during the autumn-winter seasons, the flow reverts to the east to a north-eastern direction [51, 52]. Various studies on connectivity in the southern Gulf of Mexico indicate a low east-west flow from the Campeche Bank to the Veracruz reefs; connectivity is stronger going west to east [107]. However, ocean currents in this region are complex, with eddies and seasonal shifts in direction within the inner shelf, showing fluxes from east to west during summer and west to east during winter [52]. The pattern of taxonomic distribution observed in this study suggests connectivity between the reef systems given the similarity between the sampling stations grouped under Cluster one (Red), particularly sampling stations (114, 126, and

140) which are located on the Carbonate shelf near the reef systems studied by Salas-Monreal et al. [52].

In this study, the environment was characterized by spatial processes (dbMEMs), temperature, depth, salinity, and sediment variability; however, other variables, such as concentration of organic matter or nutrients, influence the distribution of polychaetes; this could explain the low values of variance explained on the variation partitioning analysis. Therefore, further studies should explore which variables may produce higher percentages of variance explained by environmental structuring.

## Acknowledgments

Thanks are due to all the participants (students, researchers, and crew) of the research expeditions "IMCA" and "DINAMO", carried out on board the R/V "Justo Sierra" (Universidad Nacional Autónoma de México) during which the biological material used in this study was collected. We also thank all the students of the Laboratorio de Ecología y Biodiversidad de Invertebrados Marinos of the Instituto de Ciencias del Mar y Limnología (ICML), UNAM, for their valuable assistance in the processing of samples. We are grateful to Jorge Castro for improving the figures. We are grateful to the two anonymous reviewers whose comments and suggestions greatly improved our manuscript.

## Author Contributions

**Conceptualization:** Benjamín Quiroz-Martínez.

**Data curation:** Benjamín Quiroz-Martínez, Pablo Hernández-Alcántara, Vivianne Solís-Weiss, León Felipe Álvarez Sánchez.

**Formal analysis:** Benjamín Quiroz-Martínez, David Alberto Salas de León, María Adela Monreal Gómez.

**Funding acquisition:** Benjamín Quiroz-Martínez, Vivianne Solís-Weiss.

**Investigation:** Benjamín Quiroz-Martínez, Pablo Hernández-Alcántara, Vivianne Solís-Weiss.

**Methodology:** Benjamín Quiroz-Martínez, Pablo Hernández-Alcántara, David Alberto Salas de León, Vivianne Solís-Weiss, María Adela Monreal Gómez.

**Software:** David Alberto Salas de León, María Adela Monreal Gómez.

**Validation:** Benjamín Quiroz-Martínez, Pablo Hernández-Alcántara, David Alberto Salas de León, María Adela Monreal Gómez, León Felipe Álvarez Sánchez.

**Writing – original draft:** Benjamín Quiroz-Martínez.

**Writing – review & editing:** Benjamín Quiroz-Martínez, Pablo Hernández-Alcántara, David Alberto Salas de León, Vivianne Solís-Weiss, María Adela Monreal Gómez, León Felipe Álvarez Sánchez.

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
