## [Decision Letter · Decision Letter 0]

3 Jan 2024

PONE-D-23-19360Taxonomic distinctness and diversity patterns of a polychaete (Annelida) community in the continental shelf of the Southern Gulf of MexicoPLOS ONE

Dear Dr. Quiroz-Martínez,

Thank you for submitting your manuscript to PLOS ONE. After careful consideration, we feel that it has merit but does not fully meet PLOS ONE’s publication criteria as it currently stands. Therefore, we invite you to submit a revised version of the manuscript that addresses the points raised during the review process.

Dear Dr. Quiroz-Martínez,

I have received the reports from referees on your manuscript, "|Taxonomic distinctness and diversity patterns of a polychaete (Annelida) community in the continental shelf of the Southern Gulf of Mexico", submitted to Plos One.

Based on the advice received, I have decided that your manuscript needs minor changes before its publication. You must revise your manuscript following the advice of both referees and pay special attention to the data availability issue raised by referee 1.

Best regards

We look forward to receiving your revised manuscript.

Kind regards,

Marcos Rubal García, PhD

Academic Editor

PLOS ONE

Journal Requirements:

"This research was funded by the Universidad Nacional Autónoma de México DGAPA/PAPIIT Project IA202321 “Análisis y visualización de bases de datos de biodiversidad en sistemas marinos para determinar cambios en los patrones de distribución de la riqueza de especies y la abundancia de la fauna marina del Golfo de México” and through grant 628 from the Instituto de Ciencias del Mar y Limnología, Universidad Nacional Autónoma de México. We are grateful to Jorge Castro for improving the figures."

5. Please note that in order to use the direct billing option the corresponding author must be affiliated with the chosen institute. Please either amend your manuscript to change the affiliation or corresponding author, or email us at plosone@plos.org with a request to remove this option.

6. In this instance it seems there may be acceptable restrictions in place that prevent the public sharing of your minimal data. However, in line with our goal of ensuring long-term data availability to all interested researchers, PLOS’ Data Policy states that authors cannot be the sole named individuals responsible for ensuring data access (http://journals.plos.org/plosone/s/data-availability#loc-acceptable-data-sharing-methods).

7. We note that Figures 1 and 4 in your submission contain [map/satellite] images which may be copyrighted. All PLOS content is published under the Creative Commons Attribution License (CC BY 4.0), which means that the manuscript, images, and Supporting Information files will be freely available online, and any third party is permitted to access, download, copy, distribute, and use these materials in any way, even commercially, with proper attribution. For these reasons, we cannot publish previously copyrighted maps or satellite images created using proprietary data, such as Google software (Google Maps, Street View, and Earth). For more information, see our copyright guidelines: http://journals.plos.org/plosone/s/licenses-and-copyright.

      a. You may seek permission from the original copyright holder of Figures 1 and 4 to publish the content specifically under the CC BY 4.0 license. 

Additional Editor Comments :

Dear Dr. Quiroz-Martínez,

I have received the reports from referees on your manuscript, "|Taxonomic distinctness and diversity patterns of a polychaete (Annelida) community in the continental shelf of the Southern Gulf of Mexico", submitted to Plos One.

Based on the advice received, I have decided that your manuscript needs minor changes before its publication. You must revise your manuscript following the advice of both referees and pay special attention to the data availability issue raised by referee 1.

Best regards

Reviewers' comments:

Reviewer's Responses to Questions

**Comments to the Author**

1. Is the manuscript technically sound, and do the data support the conclusions?

Reviewer #1: Partly

Reviewer #2: Yes

2. Has the statistical analysis been performed appropriately and rigorously? 

Reviewer #1: Yes

Reviewer #2: Yes

3. Have the authors made all data underlying the findings in their manuscript fully available?

Reviewer #1: No

Reviewer #2: Yes

4. Is the manuscript presented in an intelligible fashion and written in standard English?

Reviewer #1: No

Reviewer #2: Yes

5. Review Comments to the Author

Reviewer #1: Please find more detailed comments in the attached annotated manuscript. This paper presents an interesting study regarding polychaete community composition in the Gulf of Mexico. The figures are well presented and the text is thorough. My main concern is the lack of openly available data. PloS One requires all data to be made available upon publication, and the authors have not provided an explanation as to why their data has not been made available. Without the publication of the data it is hard to properly review the paper. If the authors are not willing to make their data available then the manuscript should not be accepted to this journal. I believe the statistical analysis performed is robust, however some details are missing in the methods and results. I question the use of an exponential model for the data, as a linear regression may be more appropriate. There are some parts of the text which are unclear, and could benefit from reading and editing by a native english speaker - however, on the whole this does not impact the general readability of the manuscript. I suggest major revisions, depending on the authors making the data available.

Reviewer #2: A very nice paper with very strong statistical analyses. I made small changes in wording and punctuation to make it more readable.

6. PLOS authors have the option to publish the peer review history of their article (what does this mean?). If published, this will include your full peer review and any attached files.

Reviewer #1: No

Reviewer #2: No

---

## [Author Response · Author response to Decision Letter 0]

13 Mar 2024

1. We note that you have indicated that there are restrictions to data sharing for this study. PLOS only allows data to be available upon request if there are legal or ethical restrictions on sharing data publicly. For more information on unacceptable data access restrictions, please see http://journals.plos.org/plosone/s/data-availability#loc-unacceptable-data-access-restrictions.

The following information was added to the manuscript: 

Data is available from Seanoe digital repository (https://doi.org/10.17882/98735)

---

## [Decision Letter · Decision Letter 1]

10 Apr 2024

PONE-D-23-19360R1Taxonomic distinctness and diversity patterns of a polychaete (Annelida) community in the continental shelf of the Southern Gulf of MexicoPLOS ONE

Dear Dr. Quiroz-Martínez,

Thank you for submitting your manuscript to PLOS ONE. After careful consideration, we feel that it has merit but does not fully meet PLOS ONE’s publication criteria as it currently stands. Therefore, we invite you to submit a revised version of the manuscript that addresses the points raised during the review process.

The manuscript is almost ready for publication but, before final acceptance you must check the issues arisen by one referee.

We look forward to receiving your revised manuscript.

Kind regards,

Marcos Rubal García, PhD

Academic Editor

PLOS ONE

Journal Requirements:

Reviewers' comments:

Reviewer's Responses to Questions

**Comments to the Author**

1. If the authors have adequately addressed your comments raised in a previous round of review and you feel that this manuscript is now acceptable for publication, you may indicate that here to bypass the “Comments to the Author” section, enter your conflict of interest statement in the “Confidential to Editor” section, and submit your "Accept" recommendation.

Reviewer #1: (No Response)

Reviewer #2: All comments have been addressed

2. Is the manuscript technically sound, and do the data support the conclusions?

Reviewer #1: Yes

Reviewer #2: Yes

3. Has the statistical analysis been performed appropriately and rigorously? 

Reviewer #1: Yes

Reviewer #2: Yes

4. Have the authors made all data underlying the findings in their manuscript fully available?

Reviewer #1: Yes

Reviewer #2: Yes

5. Is the manuscript presented in an intelligible fashion and written in standard English?

Reviewer #1: Yes

Reviewer #2: Yes

6. Review Comments to the Author

Reviewer #1: I am happy with the changes the authors have made to the manuscript following the first round of reviews, particularly making the data publicly accessible.

My only comment concerns Figure 3, as the scale on the y-axis is incorrect. I believe the scale should increase from 0 - 600km, however the numbers shown lack the correct magnification.

Reviewer #2: (No Response)

7. PLOS authors have the option to publish the peer review history of their article (what does this mean?). If published, this will include your full peer review and any attached files.

Reviewer #1: No

Reviewer #2: No

---

## [Author Response · Author response to Decision Letter 1]

11 Apr 2024

While revising your submission, please upload your figure files to the Preflight Analysis and Conversion Engine (PACE) digital diagnostic tool, https://pacev2.apexcovantage.com/.

Figures were uploaded to PACE and approved. The PACE versions of the figures were uploaded.

References were thoroughly checked

As suggested by Reviewer 1, figure 3 was modified. 

Suggested changes to the manuscript were also included

Figures were removed from the manuscript as suggested

---

## [Editor Report · Decision Letter 2]

23 Apr 2024

Taxonomic distinctness and diversity patterns of a polychaete (Annelida) community in the continental shelf of the Southern Gulf of Mexico

PONE-D-23-19360R2

Dear Dr. Quiroz-Matínez,

We’re pleased to inform you that your manuscript has been judged scientifically suitable for publication and will be formally accepted for publication once it meets all outstanding technical requirements.

Kind regards,

Marcos Rubal García, PhD

Academic Editor

PLOS ONE
---

## [Editor Report · Acceptance letter]

26 Apr 2024

PONE-D-23-19360R2 

PLOS ONE

Dear Dr. Quiroz-Martínez, 

I'm pleased to inform you that your manuscript has been deemed suitable for publication in PLOS ONE. Congratulations! Your manuscript is now being handed over to our production team.

Kind regards, 

on behalf of

Dr. Marcos Rubal García 

Academic Editor

PLOS ONE